# Epigenetic DNA Modifications Upregulate SPRY2 in Human Colorectal Cancers

**DOI:** 10.3390/cells10102632

**Published:** 2021-10-02

**Authors:** Alexei J. Stuckel, Shuai Zeng, Zhen Lyu, Wei Zhang, Xu Zhang, Urszula Dougherty, Reba Mustafi, Qiong Zhang, Trupti Joshi, Marc Bissonnette, Samrat Roy Choudhury, Sharad Khare

**Affiliations:** 1Department of Medicine, Division of Gastroenterology and Hepatology, University of Missouri, Columbia, MO 65212, USA; ajsn6c@health.missouri.edu (A.J.S.); beefjiao@gmail.com (Q.Z.); 2Bond Life Sciences Center, University of Missouri, Columbia, MO 65201, USA; zengs@mail.missouri.edu (S.Z.); zl7w2@mail.missouri.edu (Z.L.); joshitr@health.missouri.edu (T.J.); 3Department of Electrical Engineering and Computer Science, University of Missouri, Columbia, MO 65201, USA; 4The Robert H. Lurie Comprehensive Cancer Center, Department of Preventive Medicine, Northwestern University Feinberg School of Medicine, Chicago, IL 60611, USA; wei.zhang1@northwestern.edu; 5Department of Medicine, University of Illinois, Chicago, IL 60607, USA; zhangxu@uic.edu; 6Department of Medicine, Section of Gastroenterology, Hepatology and Nutrition, The University of Chicago, Chicago, IL 60637, USA; udougher@medicine.bsd.uchicago.edu (U.D.); rmustafi@uchicago.edu (R.M.); mbissonn@medicine.bsd.uchicago.edu (M.B.); 7Institute for Data Science and Informatics, University of Missouri, Columbia, MO 65211, USA; 8Department of Health Management and Informatics, School of Medicine, University of Missouri, Columbia, MO 65212, USA; 9Department of Pediatrics, Arkansas Children’s Research Institute, University of Arkansas for Medical Sciences, Little Rock, AR 72202, USA; SRoychoudhury@uams.edu; 10Harry S. Truman Memorial Veterans’ Hospital, Columbia, MO 65201, USA

**Keywords:** colorectal cancer, DNA methylation, epigenetic regulation, SPRY2 gene expression, 5-hydroxymethylcytosine, CTCF transcription factor

## Abstract

Conventional wisdom is that Sprouty2 (SPRY2), a suppressor of Receptor Tyrosine Kinase (RTK) signaling, functions as a tumor suppressor and is downregulated in many solid tumors. We reported, for the first time, that increased expression of SPRY2 augments cancer phenotype and Epithelial-Mesenchymal-Transition (EMT) in colorectal cancer (CRC). In this report, we assessed epigenetic DNA modifications that regulate SPRY2 expression in CRC. A total of 4 loci within *SPRY2* were evaluated for 5mC using Combined Bisulfite Restriction Analysis (COBRA). Previously sequenced 5hmC nano-hmC seal data within *SPRY2* promoter and gene body were evaluated in CRC. Combined bioinformatics analyses of SPRY2 CRC transcripts by RNA-seq/microarray and 450K methyl-array data archived in The Cancer Genome Atlas (TCGA) and GEO database were performed. SPRY2 protein in CRC tumors and cells was measured by Western blotting. Increased SPRY2 mRNA was observed across several CRC datasets and increased protein expression was observed among CRC patient samples. For the first time, *SPRY2* hypomethylation was identified in adenocarcinomas in the promoter and gene body. We also revealed, for the first time, increases of 5hmC deposition in the promoter region of *SPRY2* in CRC. *SPRY2* promoter hypomethylation and increased 5hmC may play an influential role in upregulating SPRY2 in CRC.

## 1. Introduction

Sprouty (SPRY) mammalian proteins consist of 4 evolutionarily conserved family members (SPRY 1–4) that exhibit tissue-specific expression patterns, apart from the most conserved sprouty isoform SPRY2 that is ubiquitously expressed [1,2]. Sprouty proteins modulate cellular processes such as proliferation, motility, survival, and differentiation. Mechanistically, downstream mitogen-activated protein kinase (MAPK) signals are inhibited by SPRY2 as it negatively regulates receptor tyrosine kinase (RTK) signaling in response to cognate receptor ligands like vascular endothelial growth factor (VEGF) [3], platelet-derived growth factor (PDGF) [4], and fibroblast growth factor (FGF) [3]. This mechanism, however, is not uniformly observed across all cell types. For example, in some cell types, SPRY2 has been shown to induce MAPK signal activation, emphasizing its cell-specific role [5,6].

Reduced expression levels of SPRY proteins are observed in many cancers and are believed to contribute to tumorigenesis and metastasis. In many instances, SPRY2 isoform is downregulated, suggesting a possible tumor suppressor function in chronic lymphocytic leukemia CLL [7], non-small cell lung cancer NSCLC [8], hepatocellular carcinoma [9], breast cancer [10], and prostate cancer [11]. In contrast, SPRY2 is overexpressed in malignant gliomas (GBM) suggesting a potential oncogenic function in brain cancer [12]. With respect to colorectal cancer (CRC), we demonstrated, for the first time, upregulation of SPRY2 in CRC [13] and inflammatory bowel disease (IBD) [14] along with an oncogenic role of SPRY2 in CRC [15]. More recently, increased expression of SPRY2 in IBD was also confirmed by other investigators [16].

In CRC, the mechanisms regulating SPRY2 expression remain largely unknown. One study found increased SPRY2 protein, in part due to SPRY2 gene activation via β-catenin/FOXO3a binding of the *SPRY2* promoter sequence [17]. Other investigators have identified small epigenetic RNA modifiers termed microRNAs (miR-21) that negatively regulate SPRY2 expression levels through imperfect binding of the miRNA seed domain to the 3′UTR of SPRY2 transcripts [18]. Conversely, SPRY2 can regulate miRNAs. For example, we have previously demonstrated that SPRY2 negatively regulates miR-194, leading to increased AKT2 protein in colon cancer cells [15]. In this regard, several other potential epigenetic modifications of *SPRY2* have yet to be thoroughly explored in CRC, including changes in DNA 5-methylcytosine (5mC) and 5-hydroxymethylcytosine (5hmC) deposition. Indeed, *SPRY2* promoter hypermethylation is correlated with decreased SPRY2 expression levels in several cancers, including hepatocellular carcinoma [19], B-cell lymphoma [20], endometrial carcinoma [21], and prostate cancer [22]. Conversely, in mice, *SPRY2* promoter hypomethylation was associated with SPRY2 upregulation and could have implications as a driver of neuronal impairment involved in neurological diseases such as Alzheimer’s [23].

We hypothesized, for the first time, that altered 5mC and/or 5hmC DNA modifications correlate with SPRY2 expression changes in CRC. Herein, we report increased SPRY2 expression at the transcript and protein level in several CRC datasets and patient samples, respectively. In addition, upregulation of SPRY2 was found tightly correlated with decreased promoter 5mC content and increased levels of 5hmC deposition in the promoter region of *SPRY2*.

## 2. Materials and Methods

### 2.1. Human Tissue Collection

The Department of Surgical Pathology at The University of Chicago [IRB protocol (10-209-A)] provided adenocarcinomas and adjacent normal-appearing colonic mucosa samples. Care was taken to ensure that the underlying muscle and necrosis tissue was excluded from resected tumor and adjacent normal-appearing colonic mucosa. Colonocytes were isolated as previously described and Western blotting was performed to check sample purity assessed by cytokeratin20 (colonocyte marker) [24]. Patient demographic information, tumor stage, and location can be found in Appendix A.

### 2.2. Isolation of Coloncytes from Colonic Mucosa

Colonocytes were isolated following previously published methods [25,26,27]. Briefly, colon tissues were removed and minced with a razor blade into 2 mm fragments that were then placed in tubes supplemented with 6 mL sterile ice-cold transport media, 50 IU/mL penicillin (Millipore Sigma, St. Louis, MO, USA), 50 µg/mL streptomycin (Millipore Sigma, St. Louis, MO, USA) and 0.5 mg/mL gentamycin (Millipore Sigma, St. Louis, MO, USA). Care was taken to wash tissue (3×) by inversion and re-suspended in 10 mL chelating buffer (transport media plus 1 mM EDTA (Millipore Sigma, St. Louis, MO, USA) and 1 mM EGTA (Millipore Sigma, St. Louis, MO, USA). In order to release colonocytes into the supernatant, processed tissue was placed on a shaker overnight at 4 °C. Supernatants containing the colonocyte fraction were then combined.

### 2.3. Cell Culture

All colon cancer and normal colon cell lines were obtained from ATCC and cultured at 37 °C in a humified atmosphere of 95% O_2_ and 5% CO_2_ and 10% fetal calf serum I (Thermo Fisher Scientific, Waltham, MA, USA). Authentication of mycoplasma-free cell lines was confirmed by short tandem repeats by (IDEXX Bioanalytics, Columbia, MO, USA). HCT116 cells were grown in high glucose DMEM medium (Thermo Fisher Scientific, Waltham, MA, USA). RKO, Caco2, CCD-841, CCD-18Co cells were grown in ATCC-formulated Eagle’s Minimum Essential Medium (ATCC, Manassas, VA, USA). FHs74 cells were grown in an ATCC-formulated Hybri-Care medium (ATCC, Manassas, VA, USA).

### 2.4. Western Blotting

Proteins were extracted in SDS-containing Laemmli buffer, quantified by RC-DC protein assay, and subjected to Western blotting. Sample lysates and pre-stained molecular weight markers were separated by SDS-PAGE on 4–20% resolving polyacrylamide gradient gels and electroblotted to PVDF membranes. Blots were incubated overnight at 4 °C with specific primary rabbit polyclonal antibody at 1:500 dilution (SPRY2, Millipore Sigma Cat#S1444, St. Louis, MO, USA) (β-actin, Millipore Sigma Cat#A5441, St. Louis, MO, USA) (Tubulin, Millipore Sigma Cat#T6074, St. Louis, MO, St. Louis) followed by 1 h incubation with appropriate peroxidase-coupled secondary antibodies that were detected by enhanced chemiluminescence using X-OMAT film (Carestream, Rochester, NY, USA). Xerograms were digitized using Epson Perfection V600I scanner (Epson, Long Beach, CA, USA).

### 2.5. Combined Bisulfite Restriction Analysis

Genomic DNA (gDNA) was extracted and eluted in water from human tissue samples, normal colon cell lines (FHs74, CCD-18Co, CCD-841), and colorectal cancer cell lines (HCT116, RKO, Caco2) using the DNeasy Blood and Tissue Kit (Qiagen, Germantown, MD, USA). For the assays, gDNA (~200 ng) was treated with sodium bisulfite using the EZ DNA Methylation-Gold Kit (Zymo, Irvine, CA, USA) per the manufacturer’s protocol. Bisulfite-treated DNA was eluted in water. Bisulfite converted DNA (~200 ng) was used as a template under standard thermal conditions (15 min hot start 95 °C (94 °C denaturation for 30 s, 56 °C annealing temp, and 72 °C extension for 45 cycles) followed by 72 °C final extension for 10 min) in a 25 μL reaction using the PyroMark PCR kit (Qiagen, Germantown, MD, USA) according to the manufacturer’s instructions. The bisulfite-specific primers with sequences designed by MethPrimer software are shown in Appendix A [28].


Region #1: Promoter: putative CTCF binding site [2 CpGs] HpyCH4IV
Forward primer: (5′→3′)TTAAAGTTTGGGTGTGTAAGGTTAG;Reverse primer: (5′→3′)AAAACCAACAAAACACAAATATCTC;


Region #2: Promoter: upstream of the TSS [1 CpG] HpyCH4IV
Forward primer: (5′→3′)TTAGAGATGGTAAGGGAAGTGGTAT;Reverse primer: (5′→3′)AACCAAAAAAACCCAACTATTTTAC;


Region #3: Promoter: closest to the TSS [3 CpGs] BstUI
Forward primer: (5′→3′)GTAGGAGGTATTTAAAAGGGTAAT;Reverse primer: (5′→3′)AAAAAAAATTCCCTAAATCAACTC;


Region #4: Intragenic CpG island: putative CTCF binding site [4 CpGs] HpyCH4IV
Forward primer: (5′→3′)TTGTTAGGTTTTTTGTAAAGTTTTT;Reverse primer: (5′→3′)CAAATTACAAACAATACCCCCTTC;

PCR products (~7 μL) were digested in a 25 μL reaction containing either BstUI (60 °C) or HpyCH4IV (37 °C) restriction enzyme for 4 h. Restriction digests were then resolved on a 2.5% agarose gel with ethidium bromide. BstUI: cuts 5′CGCG’3 sequence and HpyCH4IV: cuts 5′ACGT’3 sequence [29]. Densitometry statistics for COBRA gel bands for *SPRY2* regions #1 and #4 in CRC patients can be found in Appendix A.

### 2.6. 5hmC-Modified Locations in the Gene Body

5hmC-Seal data were downloaded for 12 pairs of tumor (TU) and adjacent tissue (TI) samples from patients with colorectal cancer (GSE89570) [30]. In short, patient libraries were made by altering the hydroxyl group on 5hmC to contain an azide group and subsequently modified with biotin to capture enrichment of 5hmC fragments. Raw sequencing reads were summarized for the gene bodies or promoters according to the current GENCODE annotations (release 19). The normalized counts from DESeq2 by library size were log2 transformed and corrected for batch effect using linear regression. The paired t-test was used to evaluate whether the 5hmC modification levels in *SPRY2* promoter and gene body were different between tumors and adjacent tissues (*p*-value < 0.05).

### 2.7. Bioinformatics Analysis

High-throughput data from tumors from CRC patients were extracted from The Cancer Genome Atlas (TCGA) and GEO datasets (GSE8671) (GSE166427) (GSE24514), then utilized for dissecting relationships between 5mC methylation (450K microarray beta values) and mRNA gene expression (FPKM or Affymetrix mRNA expression array values) [31,32]. CRC was categorized into either respective TNM tumor sizes (T1–T4), healthy colon mucosa, adjacent to the tumor control, adenomas, adenocarcinomas, and also classified by *SPRY2* copy number variant score: Diploid (0) and copy number gain (1) for analysis. A python statistical data visualization library (Seaborn) was used to generate boxplots displaying methylation and mRNA expression distributions [33]. DNA methylation and histone&CTCF ChIP-seq data from CRC cell lines Caco2 and HCT116 were extracted from the Encyclopedia of DNA Elements (ENCODE) and analyzed in the context of SPRY2 regulation [34]. TCGA patient clinical information can be found in Appendix A. Gene-level copy number variation was downloaded and parsed by R package TCGAbiolinks [35]. The gene-level copy number variation groups were originally generated using GISTIC2 [36] and provided by The Cancer Genome Atlas (TCGA) [31]. An R statistical data visualization package (ggplot2) was used to generate the boxplot showing gene expression in different gene level copy number variation groups. The *x*-axis is gene-level copy number variation groups, which are categorized as +1 and 0, representing copy number amplification and diploid normal copy. The *y*-axis represents FPKM (mRNA gene expression) values from RNA-seq data.

## 3. Results

### 3.1. SPRY2 Is Upregulated in Adenomas and Colorectal Cancers

We evaluated SPRY2 mRNA expression in benign adenomas and matched control tissue in a publicly available dataset using the GEO2R software tool located in the National Center for Biotechnology Information (NCBI) database (GSE8671) [32]. An overall increase in SPRY2 transcripts in adenomas compared to adjacent control mucosa from matched patient samples was noted (Figure 1a). Furthermore, we also found an increase in SPRY2 transcripts in adenocarcinomas compared to adjacent control colon samples (Figure 1b,c) from two separate data sources: GEO (GSE166427) and TCGA, respectfully. Increased SPRY2 expression was even more evident in adenocarcinomas compared to that of healthy colonic mucosa (Figure 1d). Increased transcript expression of SPRY2 was also noticed in microsatellite instable tumors (MSI) compared to normal mucosa (GSE24514) (Figure 1e). In addition, we investigated whether or not copy number gains of *SPRY2* had an effect on mRNA expression in CRC patients archived in TCGA (Figure 1f). We found no group difference in SPRY2 transcript expression between CRC patients with (1) or without (0) copy number gains of *SPRY2*. To validate the observed increased transcript expression of SPRY2 in CRC across multiple datasets, protein expression in CRC patients was also assessed. We found increased SPRY2 protein expression in CRC biopsies compared to normal-appearing adjacent matched control tissue (Figure 1g). Overall, our results clearly demonstrate the early onset of SPRY2 upregulation in benign adenomas (early stage), which also continued during the stage of adenocarcinoma.

Investigations to uncover biological mechanisms regulating SPRY2 expression in CRC are scarce. Given the notable upregulation of SPRY2 mRNA and protein observed in CRC patients, it was of interest to explore the 5mC status of *SPRY2* gene, a known regulator of gene expression, and to correlate our findings with mRNA and protein abundance. According to the DNA Methylation Interactive Visualization Database (DNMIVD), *SPRY2* promoter methylation is decreased in adenocarcinomas compared to control mucosa (Figure 2a), which complements the inverse correlation found between *SPRY2* methylation and mRNA expression in adenocarcinomas archived in cBioPortal’s database (Figure 2b) [37,38].

### 3.2. CTCF Binding Sites within Transcriptional Regulatory Regions of SPRY2 Are Differentially Methylated in CRC

Only one study has investigated 5mC as a potential epigenetic regulator of SPRY2 in CRC [18]. The authors did not detect 5mC in tumor or control samples in a small promoter region (~248 bps) of *SPRY2*. To expand the scope of former studies, we aimed at exploring the abundance of 5mC in 4 putative regulatory regions of *SPRY2*, based on overlapping H3K27ac and H3K4me3 active chromatin histone markers reflected by ENCODE ChIP-seq data (hg19 source) that are displayed as signal peaks in CRC cell lines Caco2 and HCT116 (Appendix A) [34]. ENCODE also generated transcription factor ChIP-seq data, expressing putative CTCF binding sites in region #1 and region #4. The loci of interest included a putative CTCF binding site in promoter region #1, two other *SPRY2* promoter regions near the TSS (regions #2 and #3), and one small intragenic CpG island (region #4) (Appendix A). Given that 5mC can inhibit transcription factor binding, promoter region #1 of *SPRY2* was most intriguing to us as the ENCODE ChIP-seq data inversely correlated high enrichment of CTCF binding in region #1 in unmethylated Caco2 cells. This is in contrast to HCT116 cells, where *SPRY2* is highly methylated in region #1, inversely correlating with low CTCF enrichment in region #1. To our knowledge, however, the CTCF–SPRY2 axis has yet to be explored in any cancers.

In order to investigate the 5mC status of *SPRY2* in human colon cancers, DNA was extracted from 10 matched tumor and adjacent normal-appearing colonocytes and evaluated by COBRA (Figure 3a). As shown in Figure 3, each sample was resolved on two lanes: the first lane contained undigested bisulfite-treated PCR amplified DNA that served as a reference control, whereas the second lane, contained bisulfite-treated PCR amplified DNA digested with a restriction enzyme that recognizes and digests specific 5′mCpG (but not unmethylated) amplicons. A total of 7/10 colorectal tumors displayed hypomethylation in either *SPRY2* regions #1 and/or #4. Intriguingly, one tumor derived from CRC patient (#519) exhibited hypermethylation in region #1. Differential methylation was not observed in promoter regions #2 or #3. Taken together, this is the first study to our knowledge to demonstrate that putative CTCF binding regulatory regions #1 and #4 are mostly hypomethylated in colorectal tumors, which correlates with overall increased SPRY2 protein expression.

### 3.3. Increased SPRY2 5hmC Deposition Occurs in CRC

We also investigated 5hmC levels from existing 5hmC next-generation sequencing profiles in matched tumor and adjacent normal colon from CRC patients [30]. The 5hmC epigenetic marks within the promoter are known to be associated with increased transcription. We found a statistical increase of 5hmC within the promoter region of *SPRY2* in *n* = 12 tumors compared to normal colonic mucosa (Figure 3b) with read counts accumulating mostly within the last exon at the 3′ end of *SPRY2* (Figure 3c). The phenomenon of 5hmC accumulation at the 3′ end of the gene body was previously reported in several genome-wide studies in other tissues, including the liver and brain [39,40,41,42]. The authors hypothesized that 5hmC deposition may begin at the end of an open reading frame (ORFs) for a given gene undergoing active demethylation. Altogether, these observations might suggest that the combined effects of increased promoter 5hmC and decreased promoter 5mC contribute to enhanced SPRY2 expression in CRCs.

### 3.4. SPRY2 Is Differentially Methylated in CRC Cells

In parallel with tumors and adjacent normal colon samples, 3 CRC cell lines: RKO, Caco2, and HCT116 cells, 3 non-transformed cell lines: FHs74, and CCD-841 colonocytes and CCD-18Co colonic fibroblasts were assayed using COBRA in the same 4 regions of *SPRY2* as previously explored in primary malignant colonocytes and adjacent control colonocytes. In contrast to primary CRC samples, COBRA results identified 5mC accumulation spanning all 4 regions of *SPRY2* in CRC cell line HCT116, where region #1 was fully methylated and regions #2, #3, #4 were partially methylated (Figure 4a). These findings are consistent with ENCODE’s 450K array datasets for HCT116 cells, which clearly shows that all 27 CpG probes spanning the entirety of *SPRY2* are either partially or fully methylated (Appendix A). The observed 5mC levels in region #1 were much higher in RKO and HCT116 cells compared to CCD-841 and CCD-18Co cells as shown by the strong bands seen beneath the undigested PCR products in the restriction digest lanes (+) of RKO and HCT116 cells, indicating greater enzymatic digestion (ergo more 5mC) of bisulfite converted amplicons. Control cell lines FHC and FHs74 and CRC cell line Caco2 did not have detectable 5mC in any of the 4 regions (Figure 4a). 5mC modifications are likely suppressing SPRY2 in CRC cells (Figure 4b). For example, HCT116 cells exhibit very low expression levels of SPRY2 protein and concomitantly have high 5mC levels in all 4 regulatory regions. Interestingly, RKO cells contain high 5mC levels exclusively in region #1 and exhibit medium levels of SPRY2 protein, while Caco2 cells do not exhibit any form of regulatory 5mC and express the highest SPRY2 protein levels among the CRC cell lines tested (Appendix A). Thus, SPRY2 protein expression levels in CRC cell lines clearly show inverse relationships with 5mC within putative regulatory regions #1–#4 (Table 1). 

## 4. Discussion

Concerning effects on tumor growth, the precise role of SPRY2 in CRC remains controversial as one study has reported SPRY2 functions as a tumor suppressor [18], whereas our previous studies have suggested SPRY2 might function as an oncogene by promoting EMT [13,14,15]. In this respect, we demonstrated for the first time that SPRY2 increases cMet, augments growth, and favors EMT invasion phenotype. Our findings were rapidly validated by other investigators [17,43]. Remarkably, no protein-coding mutation for *SPRY2* has been reported or linked to any forms of cancer for that matter, including CRC. In this regard, we hypothesized that altered epigenetic regulation of SPRY2 is the dominant theme in CRC. In this report, we investigated two epigenetic modifications, including 5′-methylcytosine (5mC) and 5′-hydroxymethylcytosine (5hmC), and assessed whether these might correlate with changes in SPRY2 protein expression in CRC. We have demonstrated that loss in 5mC content (hypomethylation) in CTCF binding regulatory regions #1 and #4, with exception of one CRC patient that exhibited increased 5mC (hypermethylation) in region #1. Furthermore, we demonstrate that losses of 5mC within the *SPRY2* promoter inversely correlate with increased promoter 5hmC. Indeed, the phenomenon of global epigenetic dysregulation, linking decreased levels of 5mC with increased levels of 5hmC, has been observed in pontine gliomas and hematopoietic cells [44,45]. In other cases, global losses of 5hmC have been linked to increases in intragenic 5mC and subsequently decreased expression of tumor suppressor genes as reported in kidney cancer [46]. We therefore hypothesized that collectively, loss of 5mC in the *SPRY2* promoter, along with increased 5hmC within *SPRY2* promoter, might enhance SPRY2 expression in colon tumors that could promote oncogenic signals. In this regard, analysis of GEO datasets showed that SPRY2 transcripts are increased in adenomas compared to normal mucosa. We speculate that similar epigenetic aberrations observed in our adenocarcinoma samples might be occurring in the precancerous adenomas.

As a trans-regulatory element protein, CTCF may serve as a positive regulator of gene expression by promoting chromatin accessibility, enabling transcriptional activation of genes through distal enhancer-promoter interactions and/or direct binding of the promoter [47,48,49,50,51]. Irrespective of various cell types, there appear to be 32 candidate enhancer regions that may in part, regulate SPRY2 as suggested by ENCODE’s putative enhancer database tool “GeneHancer” (data not shown) [52]. We suggest a similar CTCF-dependent mechanism might regulate SPRY2 in CRC. In our hypothetical model of SPRY2 regulation, the presence of 5mC in promoter region #1 of *SPRY2* might inhibit CTCF binding, thereby leading to decreased SPRY2 protein. This is supported by the observation that in HCT116 cells *SPRY2* is methylated in all 4 regions and concomitantly SPRY2 is virtually not expressed. In contrast in RKO cells, only 1/4 regulatory regions were methylated and SPRY2 is moderately expressed, whereas in Caco2 cells, none of the 4 regions were methylated; thus, SPRY2 is highly expressed.

Several studies have identified irregular CTCF binding linked to oncogenic transcriptional misregulation in various cancer cell types. In respect to our study, one report observed that the majority of altered CTCF binding is likely due to DNA methylation modifications as opposed to sequence mutations within or around CTCF binding motifs [50]. Other investigators have also found evidence of CTCF binding inhibition by 5mC. For example, Soto-Reyes et al. demonstrated in gynecological related cancers that 5mC inhibited CTCF binding within the promoter region of tumor suppressor microRNA-125b1 (miR-125b1) [51]. In this model, CTCF promotes miR-125b transcription, and decreases in miR-125b1 transcripts were attributed to 5mC in the *miR*-*125b1* promoter inhibiting CTCF binding and gene activation.

In agreement with Feng et al., we found no evidence of 5mC or differential methylation within *SPRY2* promoter regions #2 and #3 in any colon tumors tested in this study [18]. However, when scanning 450K methylation probe data in regions #2 and #3 from TCGA (unpublished observation), we find subtle decreases in 5mC levels of dinucleotide CG#11207507 within a few base pairs of the transcriptional start site (TSS) of *SPRY2* that were associated with increased tumor sizes in CRC. As previously described in a prostate cancer model, the higher GC-rich dinucleotide percentage near the TSS of *SPRY2* renders neighboring CpGs prone to oxidative degradation by sodium bisulfite treatment, in which CG#11207507 resides [22]. This limitation hinders the ability to design competent COBRA or MSP primers within the TSS region of *SPRY2*. Thus, oxidation by sodium bisulfite-associated DNA methylation techniques like COBRA and many others prevented the evaluation of 5mC in the CpG site CG#11207507. While Feng et al. reported an inverse relationship between SPRY2 (down) and miR-21 (up) gene expression in later stages of CRC, increases in *SPRY2* promoter 5mC levels as shown in HCT116 cells and potentially some CRCs could be a consequence of miR-21 signals or could be up-stream of miR-21 gene activation or perhaps an unrelated and indirect consequence. Further studies are clearly needed to delineate the complexities of SPRY2 epigenetic regulation in CRC, especially in regard to the potential role of 5mC and 5hmC modifications of enhancer–promoter activity in SPRY2 expression. Further, in-depth analysis of *SPRY2’s* many enhancer interactions and its effects on SPRY2 gene expression is therefore warranted and are currently a focus in our laboratory.

The clinical relevance of SPRY2 research in CRC patients has just started appearing. A recent study identified a differential mRNA expression panel consisting of nine genes, in which SPRY2 was upregulated in patients with advanced rectal cancer who did not respond to preoperative chemoradiotherapy (PCRT) [53]. The authors suggest that these non-responder patients may therefore be susceptible to metastasis due to delayed tumor surgical resection and in part by increased SPRY2 expression. In this regard, SPRY2’s role in advanced rectal cancer may align with our previous reports that suggested a mechanistic function of SPRY2 as an oncogene and a positive regulator of EMT in CRC [13,15]. In addition, SPRY2 might serve as a potential biomarker with respect to anti-EGFR treatment response in colon cancers [54]. Furthermore, another recent study found that SPRY2 was increased in colon epithelial cells in patients with ulcerative colitis and Crohn’s disease [16]. More importantly, they revealed the underlying mechanism using a *SPRY2* KO mice model and found that in response to acute colitis, deletion/suppression of *SPRY2* played a protective role in maintaining the integrity of the colon epithelium against inflammation. This signaling mechanism may therefore be lost in CRC patients with constitutively expressed SPRY2, which may in part be regulated by promoter hypomethylation and increased 5hmC as discussed in our study. Furthermore, the changes in *SPRY2* 5mC and 5hmC in adenocarcinomas observed in CRC patients from this study may have clinical implications as it pertains to identifying potential adenocarcinomas that may originate from precancerous adenomas and therefore warrants further studies. In regard to previously established DNA methylation-based assays, the *SEPT9* gene promoter methylation assay was the first FDA-approved blood assay for CRC screening and early detection, displaying both high sensitivity and specificity [55].

## 5. Conclusions

In summary, apart from malignant gliomas, CRC is the second example where SPRY2 expression is increased and is responsible for augmenting cancer phenotype. Identified, for the first time, SPRY2 transcript and protein upregulation is in concordance with promoter hypomethylation and increased 5hmC in colon cancers. In the context of diagnostic strategies, this might help detect precancerous colonocytes by identifying altered SPRY2 expression or 5mC methylation patterns in region #1 and or region #4 of *SPRY2* in CRC.

## Figures and Tables

**Figure 1 cells-10-02632-f001:**
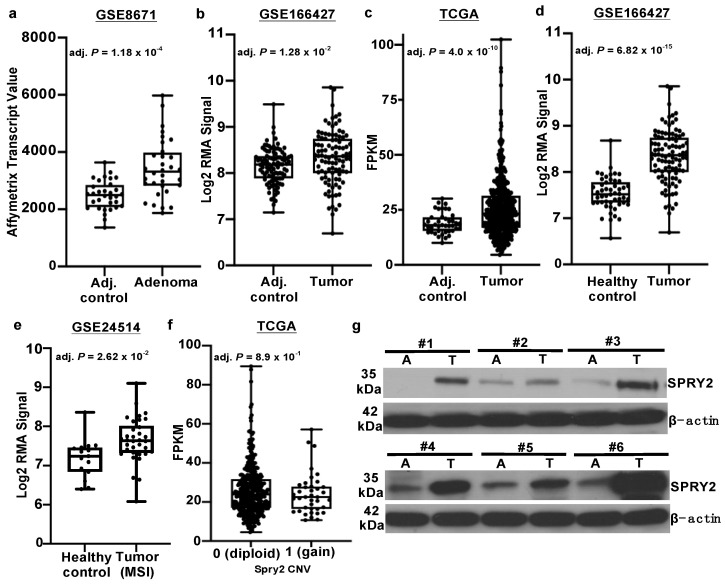
(**a**) GEO dataset (GSE8671) displaying increased SPRY2 transcripts in 32 adenomas compared to matched adjacent control colon samples. Increased SPRY2 transcript expression in adenocarcinomas compared to adjacent control from (**b**) GEO dataset (GSE166427) and (**c**) The Cancer Genome Atlas (TCGA). (**d**) Increased SPRY2 mRNA in adenocarcinomas compared to healthy mucosa (GSE166427). (**e**) Increased SPRY2 transcripts in microsatellite instable (MSI) CRC tumors compared to healthy mucosa (GSE24514) (**f**) Copy number variation score and its values representing no copy gain (0) or gain (1) of *SPRY2*, where no statistical difference in transcript expression (FPKM) was observed. (**g**) Western blot of SPRY2 showing increased expression in tumor colon (T) compared to normal colon (A) in #1-#6 CRC patients.

**Figure 2 cells-10-02632-f002:**
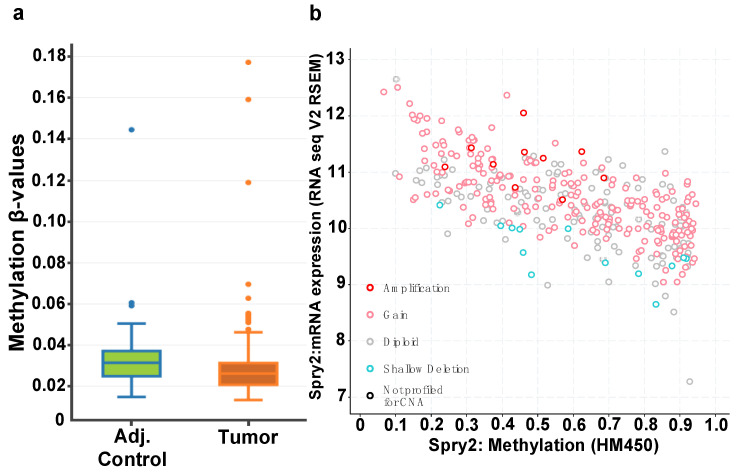
(**a**) DNMIVD figure displaying decreased promoter methylation-beta values in colon tumor compared to normal mucosa (*p*-value < 0.05). (**b**) cBioPortal’s database showcasing an inverse correlation of decreased promoter methylation-beta values with increased mRNA expression (*n* = 372) (Spearman: *p*-value = 8.21 x 10^−48^).

**Figure 3 cells-10-02632-f003:**
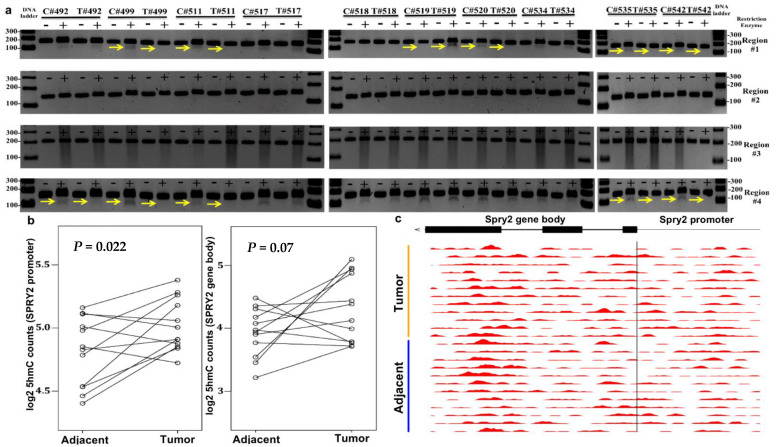
(**a**) Combined bisulfite restriction assay (COBRA) results of 4 regions of interest located in *SPRY2* from 10 colorectal cancer (CRC) patients (T(tumor)# patient ID number) and adjacent normal colonocytes (C(control)# patient ID number). Each sample was run on two lanes: lane 1) bisulfite-treated PCR amplified DNA without restriction enzyme digestion, which served as a reference control for unmethylated CpG and lane 2) bisulfite-treated PCR amplified DNA digested with a restriction enzyme recognizing amplicons containing a 5′mCpG sequence. + Lane: PCR product treated with restriction enzyme lane, − Lane: PCR product from untreated sample. Yellow arrows serve to highlight differential methylation between matched control and tumor samples for a given patient, where a visible band underneath the PCR product indicates the presence of 5mC and no visible band indicates no 5mC presence. (**b**,**c**) 5-hydroxymethylcytosine (5hmC) abundance in *SPRY2* gene bodies in colon cancers and matched adjacent mucosa (*n* = 12 samples, *p* < 0.05, paired Student’s t-test): (**b**) Increase in mean log2 fold change of 5hmC in *SPRY2* gene promoter and a marginal increase in the gene body in colon tumors (Tumor) compared to normal colon (Adjacent). (**c**) 5hmC peaks within the gene bodies of *SPRY2* in tumor and normal colon. The moving averages at 0.01 smoother span are shown. Black bars mark exons.

**Figure 4 cells-10-02632-f004:**
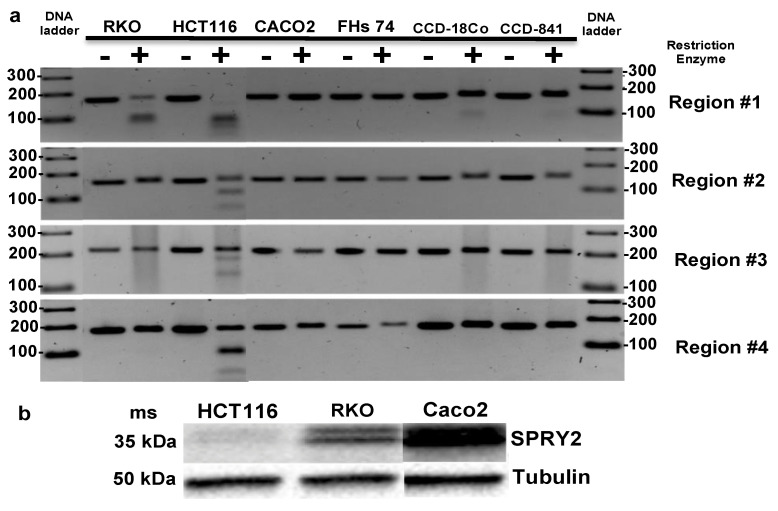
(**a**) COBRA results of 4 regions of interest located in *SPRY2* from: 2 CRC cell lines: RKO, HCT116 cells and 3 untransformed colon cell lines: FHC, FHs74, and CCD-841 and 1 fibroblast cell lines CCD-18Co cells. Region #1 shows complete PCR product digestion in HCT116 cells and regions #2, #3, and #4 show partial PCR product digestion, indicating partial methylation in regions #2–#4 and complete methylation in region #1. Region #1 also shows partial methylation in CRC cell lines: RKO and control cell lines: CCD-841 and CCD-18Co. (**b**) Western blot analysis of SPRY2 in 3 colorectal cancer cell line.

**Table 1 cells-10-02632-t001:** Inverse correlation of SPRY2 regulatory 5mC and protein expression in CRC cell lines.

	Caco2 Cells	RKO Cells	HCT116 Cells
SPRY2 protein expression	High	Medium	Low
5mC presence region #1	No	Yes	Yes
5mC presence region #2	No	No	Yes
5mC presence region #3	No	No	Yes
5mC presence region #4	No	No	Yes

## Data Availability

The data presented in this study are available on request from the corresponding author.

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
