# Peer review of "Epigenetic DNA Modifications Upregulate SPRY2 in Human Colorectal Cancers"

_cells, 2021, doi:10.3390/cells10102632_

Round 1
Reviewer 1 Report
Stuckel et al performed an in-depth analysis investigating the role of SPRY2 in CRC and identified key putative regulatory regions where their DNA methylation status could influence the expression of the gene. The authors showed a comprehensive knowledge of the gene, appropriately utilized the public data and generated new loci-specific DNA methylation and hydroxy-methylation data. Some specific comments for the manuscript are:
Supplementary Figure 1
Please include the TSS so the targeted regions or amplicons could be easily visualized relative to the gene
L150- 164
The authors should consider including genomic coordinates for the 4 regulatory regions. This will be useful for follow-up or validation studies in future.
Results section 3.1: The expression of SPRY2 seems to be highly variable in both adenomas and adenocarcinoma. Rather than saying that expression was different in tumours when compared to normal mucosa, the authors should acknowledge this huge variation in expression and provide possible reason for this huge heterogeneity?
The authors showed that SPRY2 is overexpressed in CRCS from TCGA. As the authors might be aware, genome-wide DNA methylation data from the same samples with RNA-seq data is available. It makes sense to check the methylation/expression correlation in TCGA. This could be used to further support cBIoPortal’s database shown in Figure 2?
COBRA has become very outdated. Is there a reason COBRA was chosen instead of more commonly used semi-quantitative methods e.g. Methylight, HRM, PyroMark or EpiTYPER? And the HM450K/850K seems to have many probes covering the gene?
Describe log2 5hmC count in detail (Figure 3)? Authors should have used known DNA samples with standardized hydroxy-methylation references to estimate % 5hmc levels? It is a bit hard to interpret the log2 5hmC count?
The restriction enzyme-based detection methods of DNA methylation can be temperamental? The authors could have performed the assay in technical duplicates/triplicates to reduce the potential error, especially with a small sample number here. At least samples with strong differences in methylation by COBRA could be validated by another semi-quantitative technique such as methylight or EpiTYPER (or similar)? Or check if there is any CpG probes (HM450K or HM850K) overlapping these regions?
CRC is a molecularly heterogenous disease and the SPRY2 seems to be heterogeneously expressed and methylation among CRC samples assessed in this manuscript? Is there any other patient and/or tumour characteristics that correlates with the SPRY2 expression/methylation i.e. CRCs with abnormal SPRY2 expression have something in common such as MSI, BRAF, KRAS, tumour site, grade or even survival? The authors could perform a quick association test using existing information, which will potentially add some clinical importance of the gene?
Author Response
Reviewer #1
Supplementary Figure 1: Please include the TSS so the targeted regions or amplicons could be easily visualized relative to the gene.
Response: This Suppl Fig1 is meant to provide primer information and approximate CpG density, i.e. overlapping CpG islands (in blue). Since these figures were generated by MethPrimer program, if we were to include the TSS, then this would vastly increase the number of other primers that will show up with it, which could potentially confuse the reader. Please refer to Suppl Fig 2 for the approximate location of the regions in relation to the SPRY2 TSS.
The authors should consider including genomic coordinates for the 4 regulatory regions. This will be useful for follow-up or validation studies in future.
Response: Genomic coordinates are now included in Suppl Fig 1.
Results section 3.1: The expression of SPRY2 seems to be highly variable in both adenomas and adenocarcinoma. Rather than saying that expression was different in tumours when compared to normal mucosa, the authors should acknowledge this huge variation in expression and provide possible reason for this huge heterogeneity?
Response: Because the adjusted P-values were significant based on GEO2R statistical FDR, we can state that there were group differences. However, we did provide individual data points (dots within boxplots) for each comparison to show the reader that there is a lot of overlap between control vs. tumor.
The authors showed that SPRY2 is overexpressed in CRCS from TCGA. As the authors might be aware, genome-wide DNA methylation data from the same samples with RNA-seq data is available. It makes sense to check the methylation/expression correlation in TCGA. This could be used to further support cBIoPortal’s database shown in Figure 2?
Response: Unfortunately, the CpG probes from 450K methylation data in TCGA do not cover the individual differentially methylated CpGs we tested in regions #1 and #4. We did look into regions #2 and #3 located in the CpG Island near the TSS, where the bulk of CpG probes lay and found no evidence of differential methylation between control and tumor.
COBRA has become very outdated. Is there a reason COBRA was chosen instead of more commonly used semi-quantitative methods e.g. Methylight, HRM, PyroMark or EpiTYPER? And the HM450K/850K seems to have many probes covering the gene?
Response: COBRA was used to detect large differences in methylation between control and tumor that could be visualized on a gel. This method is much cheaper and also effective in showcasing differential methylation. We plan on employing pyrosequencing to quantitate individual CpG methylation differences seen in regions #1 and #4 on a much larger patient sample scale. This will appeal to the clinical questions our lab has in regards to potential epi-biomarkers of SPRY2.
Describe log2 5hmC count in detail (Figure 3)? Authors should have used known DNA samples with standardized hydroxy-methylation references to estimate % 5hmc levels? It is a bit hard to interpret the log2 5hmC count?
Response: The read counts data were obtained from the 5hmc-Seal method, which is a pull-down based approach, i.e., enrichment of 5hmC-containing fragments. The level of 5hmC modification of a particular genomic feature (e.g., promoter, gene body) can then be represented by the count of reads, similar to RNA-seq data. The counts data were log2-transfromed so we can focus on the proportional changes (relative to absolute changes) that are more biologically relevant.
The restriction enzyme-based detection methods of DNA methylation can be temperamental? The authors could have performed the assay in technical duplicates/triplicates to reduce the potential error, especially with a small sample number here. At least samples with strong differences in methylation by COBRA could be validated by another semi-quantitative technique such as methylight or EpiTYPER (or similar)? Or check if there is any CpG probes (HM450K or HM850K) overlapping these regions?
Response: Our COBRA results in regions #1-#4 in cell lines HCT116 and Caco2 strongly correlate with ENCODE’s (Encyclopedia of DNA elements) 450K data in these regions for those cell lines. For example, ENCODE’s data revealed that SPRY2 in HCT116 cells is extensively methylated in the promoter spanning to region #4 and Caco2 is not methylated in these regions (see Suppl Fig2 under 450K probes and also Fig4a). Unfortunately, 850K data does not exist in ENCODE or TCGA datasets for colorectal patients. We will likely use pyrosequencing in the future to accurately quantitate these methylation differences.
CRC is a molecularly heterogenous disease and the SPRY2 seems to be heterogeneously expressed and methylation among CRC samples assessed in this manuscript? Is there any other patient and/or tumour characteristics that correlates with the SPRY2 expression/methylation i.e. CRCs with abnormal SPRY2 expression have something in common such as MSI, BRAF, KRAS, tumour site, grade or even survival? The authors could perform a quick association test using existing information, which will potentially add some clinical importance of the gene?
Response: We appreciate this comment. We will look into many variables in the future like the ones mentioned and also include CIMP status and other clinical markers and their potential relationship to SPRY2 expression in CRC. So far, we found no group difference in SPRY2 expression between patients with increased copies of SPRY2 and diploid patients with 2 copies of SPRY2. In fact, many patients with increased copies of SPRY2 follow an inverse correlation relating to promoter methylation and mRNA expression (Fig 2b.).
Reviewer 2 Report
In the present study, Stuckel et al., report that the up-regulation of Sprouty2 (SPRY2) in colorectal cancer (CRC) is mediated through altered 5mC and 5hmC methylation patterns. Specifically, authors provide evidence that the observed increased expression of SPRY2 is mediated through promoter and gene body SPRY2 hypomethylation, along with increased 5hmC promoter deposition.
Results are interesting, further supporting previous data from the same group, while characterize to a greater extent the underlying mechanisms that mediate, at least in part, regulation of SPRY2 expression in CRC, at an epigenetic level.
Introduction is generally well written summarizing current knowledge relevant to SPRY characteristics and function in different cancer types, including CRC, although relatively short, while results and discussion are well described and analyzed/discussed.
I have only some minor comments: Authors should pay attention, as different sections of materials and methods, particularly 2.1, 2.2 and 2.3 are quite identical to their recent paper entitled: ‘’Enhanced CXCR4 Expression Associates with Increased Gene Body 5-Hydroxymethylcytosine Modification but not Decreased Promoter Methylation in Colorectal Cancer’’ Cancers (Basel), 12(11):3104 and should be rephrased. In addition, in section 2.4 authors should provide information relevant to β-actin antibody (dilution, company), used as loading control in western blot experiments.
Author Response
Reviewer #2
Authors should pay attention, as different sections of materials and methods, particularly 2.1, 2.2 and 2.3 are quite identical to their recent paper entitled: ‘’Enhanced CXCR4 Expression Associates with Increased Gene Body 5-Hydroxymethylcytosine Modification but not Decreased Promoter Methylation in Colorectal Cancer’’ Cancers (Basel), 12(11):3104 and should be rephrased. In addition, in section 2.4 authors should provide information relevant to β-actin antibody (dilution, company), used as loading control in western blot experiments.
Response: The above methods sections were reworded and beta-actin and tubulin references were added in 2.4.